# Detection of Merkel Cell Polyomavirus (MCPyV) DNA and Transcripts in Merkel Cell Carcinoma (MCC)

**DOI:** 10.3390/pathogens12070894

**Published:** 2023-06-29

**Authors:** Sara Passerini, Carla Prezioso, Giulia Babini, Amedeo Ferlosio, Terenzio Cosio, Elena Campione, Ugo Moens, Marco Ciotti, Valeria Pietropaolo

**Affiliations:** 1Department of Public Health and Infectious Diseases, “Sapienza” University of Rome, 00185 Rome, Italy; sara.passerini@uniroma1.it (S.P.); babini.1398682@studenti.uniroma1.it (G.B.); 2Laboratory of Microbiology of Chronic-Neurodegenerative Diseases, IRCCS San Raffaele Roma, 00166 Rome, Italy; carla.prezioso@sanraffaele.it; 3Anatomic Pathology, Department of Biomedicine and Prevention, Tor Vergata University of Rome, 00133 Rome, Italy; ferlosio@med.uniroma2.it; 4Dermatologic Unit, Department of Systems Medicine, Tor Vergata University of Rome, 00133 Rome, Italy; terenziocosio@gmail.com (T.C.); elena.campione@uniroma2.it (E.C.); 5Department of Experimental Medicine, Tor Vergata University of Rome, 00133 Rome, Italy; 6Department of Medical Biology, Faculty of Health Sciences, University of Tromsø—The Arctic University of Norway, 9037 Tromsø, Norway; ugo.moens@uit.no; 7Virology Unit, Polyclinic Tor Vergata Foundation, 00133 Rome, Italy; marco.ciotti@ptvonline.it

**Keywords:** Merkel cell polyomavirus, Merkel cell carcinoma, LTAg transcript, LT truncation, integration, oncogenesis

## Abstract

Merkel cell polyomavirus (MCPyV) is the etiological agent of the majority of Merkel cell carcinoma (MCC): a rare skin tumor. To improve our understanding of the role of MCPyV in MCCs, the detection and analysis of MCPyV DNA and transcripts were performed on primary tumors and regional lymph nodes from two MCC patients: one metastatic and one non-metastatic. MCPyV-DNA was searched by a quantitative polymerase chain reaction (qPCR), followed by the amplification of a Large T Antigen (LTAg), Viral Protein 1 (VP1) and Non-Coding Control Region (NCCR). LTAg and VP1 transcripts were investigated by reverse-transcription PCR (RT-PCR). Viral integration was also studied, and full-length LTAg sequencing was performed. qPCR revealed that the primary tumor of both patients and the lymph node of one patient was positive for the small t-antigen, with an average value of 7.0 × 10^2^ copies/µg. The same samples harbored LTAg, NCCR and VP1 DNA. Sequencing results showed truncated LTAg with the conserved retinoblastoma (Rb) protein binding motif and VP1 and NCCR sequences identical to the MCC350 strain. RT-PCR detected LTAg but not VP1 transcripts. The MCPyV genome was integrated into the primary tumor of both patients. The results confirmed the connection between MCPyV and MCC, assuming integration, LTAg truncation and Rb sequestration as key players in MCPyV-mediated oncogenesis.

## 1. Introduction

Merkel cell carcinoma (MCC) is a rare but aggressive skin cancer [1] that frequently metastasizes to drain lymph nodes and distant organs [2]; therefore, a prompt diagnosis is required [3]. Since MCCs are frequently misclassified [4], today, immunohistochemistry is performed to confirm the diagnosis: a combination of cytokeratin 20 (CK20), neurofilament (NF), CK7 and thyroid transcription factor-1 (TTF-1) stains are used in order to distinguish MCC from other neoplasms [5]. In addition, other specific markers have been suggested, such as atonal homolog 1 (ATOH1) and the special AT-rich sequence-binding protein 2 (SATB2) [6].

MCC usually occurs in people with a lifelong history of intense UV exposure from the sun [7]; moreover, elderly, fair skin and immunodeficient subjects, such as HIV patients or transplant recipients, and chronic lymphocytic leukemia patients, seem to have a higher risk of developing this kind of cancer [5].

In 2008 Feng et al. isolated a new human polyomavirus from MCC called Merkel cell polyomavirus (MCPyV) and demonstrated the clonal integration of the viral genome, suggesting the possible role of this virus in MCC pathogenesis [8]. MCPyV is a small non-enveloped virus with a double-stranded DNA genome characterized by three functional domains. The early region encodes for large T (LT), small T (sT) and 57 kT antigens, and for ALTO, an alternative open reading frame from LT was identified [1]. The late region encodes for the capsid proteins, viral protein 1 (VP1) and viral protein 2 (VP2) [9] and for two mature microRNAs (miRNAs), MCV-miR-M1-5p and MCV-miR-M1-3p, which are thought to modulate viral replication [10,11]. Interposed between these two regions resides the Non-Coding Control Region (NCCR), which contains the origin of replication (Ori) and the promoters and enhancers for the regulation of early and late gene expression [1]. MCPyV NCCR polymorphisms have been described [12]; however, since the same sequence variations have been described both in MCC and non-MCC samples, no specific NCCR structure has been associated with MCC tumors [1].

To date, several studies reported MCPyV detection in MCCs, thus leading us to consider MCPyV as the major causative factor of this tumor [13]. Specifically, in almost 80% of MCC cases, viral genome integration occurs, and a truncated form of LT (tLT) is expressed [8]. tLT retains its N-terminal LXCXE motif; therefore, the ability to bind the Retinoblastoma protein (Rb) could inhibit its activity and result in cell cycle progression [14]. The C-terminus, instead of containing the DNA binding and helicase domains needed for DNA replication and cell growth inhibitory domains, is lost. Consequently, LT truncation prevents viral replication but maintains the oncogenic activity [15,16].

Based on these findings, MCPyV and UV radiation are now considered the two main etiological factors in MCC development [7].

Despite both viral positive and negative MCCs, both tumor types give similar clinical and histopathological features. However, sequencing studies have reported some differences between MCC tumors with or without the presence of MCPyV. Specifically, non-viral MCC has shown a highly damaged genome with many mutations, including the loss of function mutations in the tumor suppressor genes *RB1* and *TP53*, whereas in MCPyV-positive tumors, only a few somatic mutations have been observed [17]. In addition, a different prognosis was reported between MCPyV positive and negative MCCs. Indeed, the first one seems to show less metastatic tendencies and a better prognosis [18]. Therefore, the identification of MCPyV in MCC is important not only for diagnostic purposes but also for predicting the prognosis [19].

Considering this background and to improve our understanding of MCPyV in the pathogenesis of MCC, we investigated the presence of MCPyV DNA, its transcripts and viral DNA integration and mutations in primary tumors and regional lymph nodes from two MCC patients. Both primary tumor samples expressed a truncated LTAg, and integration of the viral genome occurred at 5q23.1 and 5q11.2, respectively. Sequence analysis showed an identity with the previously described MCC350 strain. The lymph node of the patient without metastasis was negative for MCPyV DNA, whereas the LTAg gene in the lymph node of the patient with metastasis did not contain mutations causing truncation.

## 2. Materials and Methods

### 2.1. Patients and Samples Collection

Two male patients, aged 70 and 81 years, were admitted to the Dermatology Clinic of Tor Vergata University Hospital (Rome, Italy) for the presence of a skin lesion suggestive of MCC. In the 81-year-old patient (patient 1), the lesion was localized in the left arm, while in the 70 years old patient (patient 2), the lesion was in the right thigh. In this latter patient, radiological images showed enlarged regional lymph nodes that were suggestive of metastatic involvement, whereas the lymphnode of patient 1 is tumor free.

A fresh biopsy was taken from each patient and sent to the virology laboratory for the search of MCPyV. Later on, 5 µm tick sections of formalin-fixed paraffin-embedded (FFPE) regional lymph nodes from both patients were also sent to the Virology laboratory and analyzed by PCR for the presence of MCPyV.

The study was approved by the local Ethic Committee of the University Hospital Tor Vergata (Rome, Italy) (protocol number 0015440/2019, 1 July 2019), and the patient’s informed consent was obtained.

### 2.2. Histological Diagnosis

Biopsies were fixed in 10% formalin for 24 h, and paraffin-embedded 4 μm thick serial sections were stained with Hematoxylin and Eosin for a routine histopathological examination or were employed for immunohistochemistry. Immunostaining for CK20, CK7, TTF-1, synaptophyn and Ki67 was performed using a Leica Bond-III Immunostainer (Leica Biosystems, Milan, Italy). Tissue sections were deparaffinized and subjected to heat-induced epitope retrieval for 10 min at pH 9.0 (Figure 1).

### 2.3. DNA Extraction

The total DNA was extracted from fresh and FFPE biopsies using Quick-DNA FFPE Miniprep (Zymo Research, Irvine, CA, USA), following the manufacturer’s instructions. The extracted nucleic was eluted in a final volume of 50 µL and then evaluated for its PCR suitability by amplifying the *β-globin* gene sequences [20].

### 2.4. Real Time Polymerase Chain Reaction (qPCR)

The presence and quantity of MCPyV DNA were determined by a quantitative polymerase chain reaction (qPCR) using primers and a probe for the *sT* gene sequence [21]. All samples were tested in triplicate, and the number of viral copies was calculated from standard curves, which were constructed using a ten-fold dilution series of plasmid pMCV-R17a containing the entire genome of MCPyV (Addgene, #24729) (dilution range: 10^8^–10 copies/mL). The amount of cellular DNA was quantified simultaneously using an SYBR GREEN PCR for the housekeeping of the *β-globin* gene and was used to normalize the MCPyV DNA.

### 2.5. MCPyV LTAg, NCCR and VP1 Standard PCR and Sequencing

Positive samples were subsequently amplified by a standard polymerase chain reaction (PCR) using sets of primers that were designed to detect MCPyV *LTAg* (LT1 and LT3), NCCR and *VP1* [8,22,23]. Amplification products were then analyzed by electrophoresis in 2% agarose gel stained with GelRed and observed under ultraviolet (UV) light. Moreover, in order to investigate NCCR and *VP1* sequence variations, following the purification of positive PCR products by a miPCR purification kit (Metabion, Plannegg, Germany), sequencing was performed in a dedicated facility (Bio-Fab research, Rome, Italy). The obtained DNA sequences were then aligned against the reference strain deposited in GenBank (MCC350: EU375803), using ClustalW2 on the European Molecular Biology Laboratory–European Bioinformatics Institute (EMBL–EBI) website and using default parameters (ClustalW2–Multiple Sequence Alignment) [24].

### 2.6. LT and VP1 Gene Expression

The total RNA was extracted using the Quick-RNA Miniprep Plus Kit (Zymo Research, Irvine, CA, USA). After the RNA quality and quantity assessment by A230/A260 ratios, reverse transcription was performed by a ZymoScript RT PreMix Kit (Zymo Research, Irvine, CA, USA) and, following *β-globin* gene amplification to confirm cDNA’s quality, its product was used for PCR amplification in order to determine *LT* and *VP1* gene expression [22].

### 2.7. Analysis of the MCPyV Integration Sites

The integration sites of MCPyV were examined by the detection of the integrated papilloma sequence (DIPS)–PCR technique [25] which allowed for the amplification of the junctions between viral and cellular genomes [26]. After DNA digestion with the TaqI restriction enzyme, the obtained DNA fragments were ligated to enzyme-specific adaptors and then subjected to a PCR amplification using viral- and adaptor-specific primers. The PCR products were purified and sequenced. The integration sites were defined by submitting sequences to the databases of the National Center for Biotechnology Information and analyzing them with the Basic Local Alignment Search Tool (BLAST) for genomic localization [22].

### 2.8. Sequencing Analysis of the MCPyV LT Gene

The DNA sequences from nucleotide positions 151 to 3102 (GenBank strain EU375803), corresponding to the entire *LT*, were analyzed by a PCR using a different combination of six primer sets [22]. The amplified products were then subjected to a direct sequence analysis.

## 3. Results

### 3.1. Detection of MCPyV DNA by Real-Time PCR (qPCR) and Standard PCR

MCPyV sT DNA was detected in the skin biopsies of both patients and in the lymph node belonging to patient 2 with the diagnosis of metastasis and with a viral load mean value of 7 × 10^2^ copies/µg (range 5.35–8.10 × 10^2^ copies/μg). The same samples were further positive for LT1 and LT3 amplification as well as for MCPyV NCCR and VP1 (Table 1). Remarkedly, no viral DNA was detected in the lymph node of patient 1 with any of the primers used.

### 3.2. MCPyV NCCR and VP1 Sequence Analysis

The sequencing analysis of amplified NCCRs, spanning from the nucleotide position 5077 to 5280, revealed a canonical structure in all the analyzed samples, which were identical with the reference sequence of the prototype North American MCC350, strain EU375803. The same was observed for the VP1 DNA sequence (Table 1).

### 3.3. Expression of LTAg and VP1 Transcripts

MCPyV LT (nucleotide positions 910–1152) and VP1 (nucleotide positions 3786–4137) transcripts analysis showed only the expression of the *LT* gene, whereas no VP1 transcripts were detected (Table 1).

### 3.4. Integration Analysis of MCPyV

Even if the quantity and quality of the DNA were critical for DIPS–PCR analysis, DNA from the 3 MCPyV-DNA-positive samples were suitable for this analysis. The integration site was identified to the two MCC primary tumors: in the two cases, the virus–host junction was located at nucleotide position 2738 and 2597 of the MCPyV *LT* gene, respectively, and viral DNA sequences were inserted into the long arm of chromosome 5. (Table 2). No integration of viral DNA was detected in the lymph node of patient 2.

### 3.5. DNA Sequencing Analysis of the MCPyV LT Gene

A sequence analysis of the full-length *LT* gene (nucleotide positions 151-3102) showed that MCPyV strains found in our MCPyV positive samples were consistent with the one of the MCC350, EU375803 strain and conserved the pRb-binding motif LFCDE found in LTAg of the other MCPyV strain, with the exception of a wild type non-tumor derived one, which had the motif LFCDK (JN038578; [15]).

The *LT* gene found in the two primary tumors displayed frameshift mutations, which arose from deletions and generated stop codons; therefore, a truncated LT was observed. Specifically, these mutations were localized downstream from the Rb-binding site and led to the truncation of the exon 2 encoding the helicase domain. The full-length LT detected in the lymph node sample, instead, did not reveal mutations causing stop codons at the C-terminus, although some mutations, resulting in amino acid substitutions, were observed.

## 4. Discussion

Our results confirm the well-known association between MCPyV and MCC. Indeed, MCPyV DNA was found in two MCC tumor biopsies and in the lymph node of a patient with the diagnosis of metastasis by five different primer sets (sT, LT1, LT3, NCCR and VP1). Moreover, in order to improve knowledge about NCCR and VP1 variability in MCPyV strains in MCC tumors, sequence analysis was performed. All analyzed NCCRs showed a complete identity with the MCC350 strain (EU375803), thus supporting the fact that no specific variations are associated with MCC [1]. The same was observed for VP1 sequences.

Since MCPyV DNA detection is not sufficient to define the MCPyV role in MCC, LT and VP1 gene expression was investigated. As reported in most MCPyV-positive MCCs, LT but not the VP1 gene was detectable at the RNA level, suggesting that viral replication was hampered, which is common in virus-mediated oncogenesis [27].

MCPyV integration into the host genome is considered a key factor in tumorigenesis; therefore, the MCPyV genome status was also examined. The MCPyV integration sites were identified in MCC primary tumors but not in the metastatic lymph node. The viral genome was inserted into the long arm of chromosome 5, which was reported as the most frequent localization [28]. The integration sites occurred downstream from the Rb-binding domain, within exon 2 of the *LT* gene, supporting the fact that the integrated genome was harbored at the 3′ ends of the *LT* gene [25,29]. In addition, in primary tumors, the sequence analysis of the full-length *LT* gene showed frameshift mutations that generated a truncated oncoprotein that retained the Rb binding domain but lacked the helicase one. This truncated protein preserved the transformation ability by Rb sequestration; however, the viral replication capacity was lost, as demonstrated by the absence of VP1 gene expression. Despite previous studies reporting the same molecular signature in metastases and primary tumors [25], in our study, the metastatic lymph node did not show integration, and the full-length LT gene sequence was amplified. This finding might suggest that, in this case, integration and LT truncation could be a late event in MCPyV-induced oncogenesis, strengthening the assumption that an alternative mechanism involving Rb sequestration or the action of sT is required. Another explanation for the discrepancy between the viral genome state in the primary tumor and the metastatic lymph node could be the presence of combined MCC, which occurs in 5–20% of all MCC cases [30,31]. Combined MCC with SCC in lymph node metastasis has been described [32,33,34]. Further studies are required to support this hypothesis.

Indeed, besides integration, MCPyV-mediated oncogenesis involves LT’s capability to bind the tumor-suppressor protein Rb by its LXCXE motif [15]. In our MCPyV-positive samples (primary tumors and metastatic lymph node), the *LT* gene sequence analysis revealed that, even when LT truncation occurred, this motif was conserved, supporting its potential role in MCC development. Moreover, sT is known to be implicated in tumorigenesis since it is able to activate several oncogenic signaling pathways [35]. As reported by previous studies [36], sT’s transforming capacity is independent of LT expression; therefore, it could be able to stimulate tumor progression even when LT is not directly involved.

## 5. Conclusions

This study contributes to strongly supporting MCPyV’s role in MCC pathogenesis, assuming viral integration and the expression of a truncated LT gene with a preserved LXCXE motif, which is able to bind and sequestrate with the Rb protein as a crucial factor in MCPyV-mediated oncogenesis.

## Figures and Tables

**Figure 1 pathogens-12-00894-f001:**
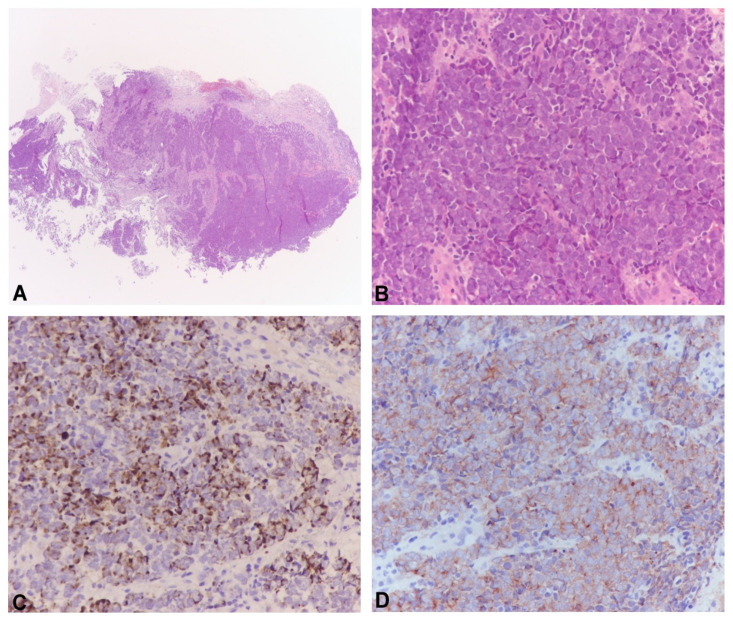
Representative pictures of Merkel cell carcinoma. At a low magnification, nodular dermal-hypodermic neoplasia with small round blue cell appearance and crush artifact (**A**). At higher magnification, the neoplastic cells showed finely dispersed chromatin, indistinct nucleoli and scant cytoplasm with mitotic figures and apoptosis (**B**). The immunohistochemical study demonstrated characteristic CK20 (**C**) and synaptophysin “hot-dots” positivity (**D**), confirming the diagnosis. (**A**,**B**) Hematoxilyn-Eosin stain; original magnification: 20× for (**A**) and 200× for (**B**–**D**).

**Table 1 pathogens-12-00894-t001:** Summary of MCPyV viral load, DNA sequences and LTAg/VP1 transcripts detected in MCC patients.

*Patient*	*Age, Gender*	*Site*	*β-Globin DNA*	*Viral Load (gEq/mL)*	*LT*		*NCCR*	*VP1*	*β-Globin cDNA*	*LT*	*VP1*
					*LT1*	*LT3*				*Transcript*	*Transcript*
1	81, M	Left arm	+	8.10 × 10^2^	+	+	canonical	canonical	+	+	-
		Lymph node	+	-	-	-	-	-	+	-	-
2	70, M	Right thigh	+	7.50 × 10^2^	+	+	canonical	canonical	+	+	-
		Lymph node	+	5.35 × 10^2^	+	+	canonical	canonical	+	+	-

**Table 2 pathogens-12-00894-t002:** Integration analysis of MCPyV in MCC primary tumors.

*Patients*	*Viral Junction*	*Cellular Junction*
1	5′-2738 (LT)	5q23.1
2	5′-2597 (LT)	5q11.2

## Data Availability

The datasets containing all data analyzed, supporting the results of this study, will be made available by the authors, without undue reservation.

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
