# Peer review of "Detection of Merkel Cell Polyomavirus (MCPyV) DNA and Transcripts in Merkel Cell Carcinoma (MCC)"

_pathogens, 2023, doi:10.3390/pathogens12070894_

Round 1
Reviewer 1 Report
The manuscript “Detection of Merkel Cell Polyomavirus (MCPyV) DNA and transcripts in Merkel Cell Carcinoma (MCC)” is a very interesting communication of two patients presenting with nodally metastasized MCCs.
A few comments:
Abstract
Line 1: please add “… the majority ...” of MCC. As the authors state correctly in the manuscript, there are approx. 80% of MCCs associated with MCPyV and 20% with UV radiation.
Introduction
Page 2, line 91 – 94: “Considering this background and to improve our understanding of MCPyV in the pathogenesis of MCC, we investigated the presence of MCPyV DNA, its transcripts and viral DNA integration and mutations in primary tumors and regional lymph nodes from two MCC patients.”:
It is always discussable if an introduction needs to summarize the aim of the study at all. Since the authors decided to do so, they should address the novelty of this communication. Although every single case of MCC on which research is performed and later published is an added value for data generation and of contributing in one or the other sense to an improvement of our understanding. However, all the tests have previously been applied to MCCs in previous studies by others. As far as I have understood this communication the unique selling point of interest for the readership is the combination of all the techniques on two metastasized MCCs. I suppose that this communication would benefit if this was mentioned here.
Materials and Methods
Page 4, line 132-133: please indicate exactly in Materials and Methods which test has been performed on FFPE- or fresh frozen tissues. Please also add to Table 1 the control data for each sample tested, i.e., ß-globin for RT-PCR and probably the results of SDS ladder (or others) for the DNA PCR.
Page 4, line 159 – 164: please explain why did the authors did not use a MCPyV RNA situ hybridization to test MCPyV transcript expression in the tumor cells? Does the ZymoScript RT PreMix Kit contain a DNAse digestion prior to reverse transcription?
Results & Discussion
The finding that the metastasis of patient #1 is MCPyV-negative in all tests is remarkable. Especially in the context that the authors show integration of MCPyV in the primaries. This would mean that the MCCs harboring integrated MCPyV lost MCPyV in the lymph node metastasis. In my view this finding deserves more attention:
- Please clearly state in the result text that the nodal metastasis of patient #1 is completely MCPyV-negative.
-> Although the loss of integrated MCPyV in both patients can be considered, the most likely needs to be excluded, i.e., please confirm for both cases that the nodal metastases indeed originate from the MCPyV-positive MCCs by MSI DNA profiling. Or is there an UV signature in the metastasis of patient #1? A metastasis of a MCPyV-negative MCC or another neuroendocrine small cell carcinoma needs to be excluded.
-> A MCPyV DNA- and RNA-in situ hybridization would significantly contribute to the understanding of the molecular findings.
Page 5, line 209 – 218 & discussion:
· Please comment on the integration of MCPyV! Full length? Concatemers? Next to the cited references there are more studies to investigate the “at random” integration of MCPyV.
The conclusion of the authors “This finding suggests that, in this case, integration and LT truncation could be a late event in MCPyV-induced oncogenesis,..” is not sufficiently supported by the data in this communication.
Author Response
Reviewer 1
The manuscript “Detection of Merkel Cell Polyomavirus (MCPyV) DNA and transcripts in Merkel Cell Carcinoma (MCC)” is a very interesting communication of two patients presenting with nodally metastasized MCCs.
A few comments:
Abstract
Line 1: please add “… the majority ...” of MCC. As the authors state correctly in the manuscript, there are approx. 80% of MCCs associated with MCPyV and 20% with UV radiation.
Thank you. We agree with the reviewer and have changed the first sentence of the abstract into:
Merkel Cell Polyomavirus (MCPyV) is the etiological agent of the majority of Merkel cell carcinoma (MCC), a rare skin tumor.
Introduction
Page 2, line 91 – 94: “Considering this background and to improve our understanding of MCPyV in the pathogenesis of MCC, we investigated the presence of MCPyV DNA, its transcripts and viral DNA integration and mutations in primary tumors and regional lymph nodes from two MCC patients.”:
It is always discussable if an introduction needs to summarize the aim of the study at all. Since the authors decided to do so, they should address the novelty of this communication. Although every single case of MCC on which research is performed and later published is an added value for data generation and of contributing in one or the other sense to an improvement of our understanding. However, all the tests have previously been applied to MCCs in previous studies by others. As far as I have understood this communication the unique selling point of interest for the readership is the combination of all the techniques on two metastasized MCCs. I suppose that this communication would benefit if this was mentioned here.
Thank you for your suggestion. We agree with the reviewer that the test and methods used in our study are commonly used. However, many of the reports describing the presence of MCPyV DNA in MCC, do not always examine the expression of viral transcripts, nor are integration site identified, viral genome sequences determined, and metastasized cells investigated. Our study investigated all these parameters. Since the introduction contains the aim of the study, we have also included a summary of our findings by adding following text:
Both primary tumor samples expressed a truncated LTAg and integration of the viral genome occurred at 5q23.1 and 5q11.2, respectively. Sequence analysis showed identity with the previously described MCC350 strain. The lymph node of the patient without metastasis was negative for MCPyV DNA, whereas the LTAg gene in the lymph node of the patient with metastasis did not contain mutations causing truncation.
This is the corrected version as in the revised manuscript.
Materials and Methods
Page 4, line 132-133: please indicate exactly in Materials and Methods which test has been performed on FFPE- or fresh frozen tissues. Please also add to Table 1 the control data for each sample tested, i.e., ß-globin for RT-PCR and probably the results of SDS ladder (or others) for the DNA PCR.
Page 4, line 159 – 164: please explain why did the authors did not use a MCPyV RNA situ hybridization to test MCPyV transcript expression in the tumor cells? Does the ZymoScript RT PreMix Kit contain a DNAse digestion prior to reverse transcription?
Thank you for your observations. Following DNA extraction performed by a commercial kit (Quick-DNA FFPE Miniprep, Zymo Research, Irvine, CA), specific for biopsies, all tests were carried out on both FFPE and fresh frozen tissues. The control data have been added to Table 1, specifically, ß-globin for both DNA and RT-PCR.
The expression of MCPyV transcripts was tested by RT-PCR, following previously published protocols (21385344, 23322199). Moreover, we decided to perform RT-PCR since it is more sensitive than RNA in situ hybridization (PMID 15297527). The ZymoScript RT PreMix Kit does not contain a DNAse digestion prior to reverse transcription. According to the manufacturer’s instructions, after the RNA sample is added to ZymoScript RT PreMix, the reaction is incubated for 2 min at 25 °C to initiate the reverse-transcription step; then the extension phase occurred at 25 °C for 10 min. Finally, inactivation of the RT enzyme is performed at 95 °C for 1 min.
Results & Discussion
The finding that the metastasis of patient #1 is MCPyV-negative in all tests is remarkable. Especially in the context that the authors show integration of MCPyV in the primaries. This would mean that the MCCs harboring integrated MCPyV lost MCPyV in the lymph node metastasis. In my view this finding deserves more attention:
- Please clearly state in the result text that the nodal metastasis of patient #1 is completely MCPyV-negative.
Thank you. We have added following text (line 193-194 in the revised manuscript):
No viral DNA was detected in the lymph node of patient 1 without metastasis with any of the primers used.
This is the corrected version as in the revised manuscript.
-> Although the loss of integrated MCPyV in both patients can be considered, the most likely needs to be excluded, i.e., please confirm for both cases that the nodal metastases indeed originate from the MCPyV-positive MCCs by MSI DNA profiling. Or is there an UV signature in the metastasis of patient #1? A metastasis of a MCPyV-negative MCC or another neuroendocrine small cell carcinoma needs to be excluded.
Thank you. In our study, the patient #1 was free from metastasis and, as we reported in Table 1, the lymph node was MCPyV negative. Only in the patient #2, instead, a nodal metastasis was observed and MCPyV was found both in the primary tumor and in the lymph node, supporting that the metastasis originates from the MCPyV-positive MCC.
Anyhow, if a MCPyV negative metastasis would be observed, we could suppose the presence of a combined MCC. Indeed, combined MCC (mostly with small squamous carcinoma) occurs in 5-20% of all MCCs (PMID:36075957; 36766554). Combined MCC with SCC with lymph node metastasis have been described (28570390, 33533503, 36377830).
We added this sentence to the revised version:
Another explanation for the discrepancy of the viral genome state in the primary tumor and the metastatic lymph node could be the presence of combined MCC, which occurs in 5-20% of all MCC cases [30,31]. Combined MCC with SCC with lymph node metastasis have been described [32-34].
-> A MCPyV DNA- and RNA-in situ hybridization would significantly contribute to the understanding of the molecular findings.
Thank you for your suggestion. However, since we do not have sufficient material, we are not able to perform DNA and RNA-in situ hybridization. Moreover, as we mentioned above, we followed previously published protocols which revealed to be appropriate to study the state of viral genome as well as the expression of viral transcripts.
Page 5, line 209 – 218 & discussion:
Please comment on the integration of MCPyV! Full length? Concatemers?
Thank you. The protocol we followed for the analysis of the integration sites do not allow us to determine whether the integrated MCPyV genome might be present as concatemer.
Next to the cited references there are more studies to investigate the “at random” integration of MCPyV.
Thank you. Integration site was investigated in MCC cell lines and primary tumors with their metastasis. A random integration was found (30873613, 32833988), although other studies suggest some preference for chromosome 5 (26569308, 32878339). However, more integration sites need to be determined to establish whether integration is at random or whether there are hotspot sites.
The conclusion of the authors “This finding suggests that, in this case, integration and LT truncation could be a late event in MCPyV-induced oncogenesis,..” is not sufficiently supported by the data in this communication.
We have changed to: “This finding might suggest that, in this case, integration and LT truncation could be a late event in MCPyV-induced oncogenesis…”
And added the sentence: “Further studies are required to support this hypothesis.”

Reviewer 2 Report
The manuscript by Passerini, Pietropaolo et al., is a short communication describing the molecular findings from two cases of the rare Merkel Cell Carcinoma, a malignant and very aggressive tumor of the skin, which has been associated with the presence of a Polyomavirus (MCPyV).
The authors demonstrate the presence and integration of Merkel Cell Polyomavirus (MCPyV) in the two samples. They also show the sequencing and structural analysis of the gene encoding for the main oncogenic protein, Large T-Antigen from theses cases, demonstrating that the RB binding motif is intact, while the gene is truncated.
The work is scientifically sound, well designed, presented in logical order and with adequate analysis. The results presented are clear, concise and compelling. These results are novel, and definitely further the knowledge on the physiopathology and this recently newly discovered Polyomavirus, and will be of general interest to the scientific community, but especially to viral oncologists.
The only suggestion I do have to improve the manuscript is to add perform immunohistochemustry on these two samples for the Merkel Cell Polyomavirus specific large T-Antigen, and adding a picture of the result. Actual expression of T-Antigen would be important to correlate with the present of the gene, and if not expressed, that would also be very interesting data.
Author Response
Reviewer 2
The manuscript by Passerini, Pietropaolo et al., is a short communication describing the molecular findings from two cases of the rare Merkel Cell Carcinoma, a malignant and very aggressive tumor of the skin, which has been associated with the presence of a Polyomavirus (MCPyV).
The authors demonstrate the presence and integration of Merkel Cell Polyomavirus (MCPyV) in the two samples. They also show the sequencing and structural analysis of the gene encoding for the main oncogenic protein, Large T-Antigen from theses cases, demonstrating that the RB binding motif is intact, while the gene is truncated.
The work is scientifically sound, well designed, presented in logical order and with adequate analysis. The results presented are clear, concise and compelling. These results are novel, and definitely further the knowledge on the physiopathology and this recently newly discovered Polyomavirus, and will be of general interest to the scientific community, but especially to viral oncologists.
The only suggestion I do have to improve the manuscript is to add perform immunohistochemustry on these two samples for the Merkel Cell Polyomavirus specific large T-Antigen, and adding a picture of the result. Actual expression of T-Antigen would be important to correlate with the present of the gene, and if not expressed, that would also be very interesting data.
Thank you for your suggestion. Many of the reports describing the presence of MCPyV DNA in MCC and performing immunohistochemistry, do not always examine the expression of viral transcripts, nor are integration site identified, viral genome sequences determined, and metastasized cells investigated. The novelty of our study was that we investigated all these parameters. Consequently we did not have sufficient material to detect LTAg by immunohistochemistry. We will consider your suggestion for our next investigations, asking for more material.

Reviewer 3 Report
In this manuscript, Passerini et al. present their molecular analysis of two Merkel Cell Carcinoma (MCC) cases and show that in both cases, the MCPyV genome is chromosomally integrated into the tumour cells and the LTag gene is transcribed, while VP1 is not. Interestingly, MCPyV was not present in a chromosomally integrated form in the lymph node. The study of this rare tumour confirms previous investigations that integration of viral DNA often occurs in MCC and that human chromosome 5 is usually the integration site. In addition, the sequence analyses show the absence of NCCR mutations, supporting the conclusion that these are not required for tumour induction.
The manuscript is clearly written, and the methods are appropriate and sound. The only question I have is related to the MCPyV-positive lymph node. I find it hard to understand how it is possible that the genome is not integrated into the presumably present metastatic cells in the lymph node. I find the suggested explanation that it is a late event unconvincing. Do the findings not suggest a heterogenous primary tumour in which cells with and without integrated MCPyV genome are present? Alternatively, could integration not be picked up in the lymph node due to mutations or other reasons? Please clarify.
line 249 "recurrent in virus mediated oncogenesis". I don't think the word recurrent is appropriate. Is 'common' meant?
Author Response
Reviewer 3
In this manuscript, Passerini et al. present their molecular analysis of two Merkel Cell Carcinoma (MCC) cases and show that in both cases, the MCPyV genome is chromosomally integrated into the tumour cells and the LTag gene is transcribed, while VP1 is not. Interestingly, MCPyV was not present in a chromosomally integrated form in the lymph node. The study of this rare tumour confirms previous investigations that integration of viral DNA often occurs in MCC and that human chromosome 5 is usually the integration site. In addition, the sequence analyses show the absence of NCCR mutations, supporting the conclusion that these are not required for tumour induction.
The manuscript is clearly written, and the methods are appropriate and sound. The only question I have is related to the MCPyV-positive lymph node. I find it hard to understand how it is possible that the genome is not integrated into the presumably present metastatic cells in the lymph node. I find the suggested explanation that it is a late event unconvincing. Do the findings not suggest a heterogenous primary tumour in which cells with and without integrated MCPyV genome are present? Alternatively, could integration not be picked up in the lymph node due to mutations or other reasons? Please clarify.
Thank you. The hypothesis of a heterogeneous primary tumor containing both integrated and episomal cells could be considered. Previous studies suggest the co-existence in the DNA of MCC-tumor biopsies of both episomal form and truncated integrated form of MCPyV (22342276). Moreover, there is one PloS One paper claiming to find virus particles in a virus-positive MCC, suggesting non-integrated viral DNA (19305499). However, further studies are warranted to better understand this hypothesis.
line 249 "recurrent in virus mediated oncogenesis". I don't think the word recurrent is appropriate. Is 'common' meant?
Thank you. We agree with the reviewer and have replaced the word ‘recurrent’ with ‘common.

Round 2
Reviewer 1 Report
I would like to thank the authors that they have addressed all points and I think that the manuscript has improved substantially.
One minor point for possible improvement:
Please state somewhere in manuscript that the lymphnode of patient #1 is tumor free. As the authors could see in my previous comments that I thought that this lymphnode was tumor positive...
And one personal opinon point:
Although, I am still not convinced that the data presented in this study provide sufficient evidence that "in this case, integration and LT truncation could be a late event in MCPyV-induced oncogenesis…”, I do accept and appreciate the modifications made by the authors concerning this topic and I sure do not want to delay the publishing of this manuscript.